# Acute Effect of Upper-Lower Body Super-Set vs. Traditional-Set Configurations on Bar Execution Velocity and Volume

**DOI:** 10.3390/sports10070110

**Published:** 2022-07-14

**Authors:** Guillermo Peña García-Orea, David Rodríguez-Rosell, Daniel Segarra-Carrillo, Marzo Edir Da Silva-Grigoletto, Noelia Belando-Pedreño

**Affiliations:** 1Department of Physical Activity and Sport, University of Murcia, 30003 Murcia, Spain; 2Physical Performance & Sports Research Center, Pablo de Olavide University, 41013 Seville, Spain; davidrodriguezrosell@gmail.com; 3Department of Sport Science, Miguel Hernández University, 03202 Alicante, Spain; dani.segarra07@gmail.com; 4Department of Physical Education, Federal University of Sergipe, São Cristóvão 49100, Brazil; medg@ufs.br; 5Department of Physical Activity and Sport Science, European University of Madrid, 28670 Madrid, Spain; noelia.belando@universidadeuropea.es

**Keywords:** set configuration, neuromuscular performance, velocity-based training, resistance training

## Abstract

This study aimed to compare the effect on bar execution velocity and number of repetitions between two velocity-based resistance training protocols only differing in the set configuration of the full-squat (SQ) and bench-press (BP) exercises. Moderately strength-trained men were assigned to a traditional (TS, *n* = 9)- or an alternating-set (AS, *n* = 10) configuration group to perform four testing sessions against different relative loads (55–60–65–70% 1RM). Relative load, magnitude of intra-set velocity loss (%VL), number of sets, inter-set recovery time, and exercise order were matched for both groups in each session. Mean propulsive velocity of the first repetition (MPV_first_), average number of repetitions per set (NRS), total number of repetitions (TNR), and total training time per session (TT) were measured. No significant differences between training conditions were observed for any relative load in MPV_first_, NRS, and TNR in both exercises. The TS group completed a significantly higher number of repetitions (*p* < 0.05) at faster velocities (MPV > 0.9–1.1 m·s^−1^) in the SQ. In conclusion, training sessions performing AS between SQ and BP exercises with moderate relative loads and %VL result in similar bar execution velocity and volume, but in a more time-efficient manner, than the traditional approach.

## 1. Introduction

Traditionally, resistance training (RT) sessions are conducted using 2 to 5 min inter-set recovery intervals and multiple sets of lower- and upper-body exercises [1]. These recovery periods minimize decreases in volume (number of repetitions) and intensity between sets but result in long, time-consuming workouts [2]. In this regard, several time-saving training techniques have been previously examined where sets are performed alternately between different exercises (usually two), either targeting the same agonist muscle group (e.g., super-set training) or involving antagonistic muscle actions (e.g., agonist–antagonist paired set) [3,4,5,6,7,8,9]. During so-called paired-set training, exercises involving agonist–antagonist muscles are performed alternately with a limited rest period or without a rest period between sets [10]. Therefore, this training set configuration differs from the traditional set, where all sets of the same exercise are performed before the execution of all sets of the next exercise [9]. In summary, these studies showed that agonist–antagonist paired-set training could allow (a) a substantial reduction in training time for an equivalent training volume (i.e., number of sets), and (b) the performance of similar or higher total repetitions per set compared to the traditional configuration [3,4,5,6,7,8,9]. However, it is also likely that some of these time-saving strategies during RT may result in significant acute neuromuscular performance impairment since the accumulated residual fatigue could reduce the capacity to continue applying force [2]. As a consequence, this fact may compromise strength gain adaptations, although knowledge about the chronic effect of performing paired sets on neuromuscular function following a training program/intervention is scarcer [3,11].

On the other hand, paired exercises alternating upper- and lower-body muscle groups performed successively (i.e., paired alternating-limb sets) have rarely been considered [2,12,13]. Some evidence has highlighted that during these time-efficient workout configurations, upper-body multi-joint exercises performed during lower-body exercises rest intervals could induce decreases in both the number of repetitions to failure and repetition power output [2]. In fact, most of the research in this matter has considered volume load (i.e., total repetitions × kilograms lifted) as a criterion or determinant variable of neuromuscular performance and training efficacy [5,7,8,9,14]. However, it is not clear if any acute neuromuscular performance impairment (e.g., repetition movement velocity or number of repetitions per set) may occur when paired alternating-limb sets (e.g., full squat and bench press) are performed using repetitions per set without reaching muscle failure. This issue is especially relevant considering that repetitions to failure have been questioned to be necessary to promote additional neuromuscular improvements and may even impair strength development at high velocities [15,16]. In this regard, velocity-based RT (VBRT) has been recognized as a highly effective and reliable methodology for training prescription and load monitoring during RT programs [17]. In addition, monitoring execution velocity loss during the set has been reported to be a reliable indicator of the degree of fatigue that is incurred during RT sessions and an accurate variable to prescribe training volume [18].

Accordingly, considering the above concerns, it is required to elucidate the acute response of paired sets performing repetitions well ahead of reaching muscle failure and implying opposite limbs (i.e., upper- and lower-body muscle groups) on neuromuscular performance (e.g., bar velocity). Thus, the aim of the present study was to compare the effect on bar execution velocity and volume (i.e., repetitions per set) of traditional vs. alternating-set configurations in the full-squat (SQ) and bench-press (BP) exercises using a VBRT approach. In this way, we hypothesize that performing alternating sets between SQ and BP exercises will not affect execution velocity and repetitions per set compared to traditional sets if a moderate degree of fatigue in the set is induced.

## 2. Materials and Methods

### 2.1. Study Design

A cross-over research design was used to compare the effect on execution velocity and volume between two VBRT protocols only differing in the set configuration (structuring) of the SQ and BP exercises. The traditional-set (TS) group first performed all SQ sets and subsequently all BP sets, and the alternating-set (AS) group performed SQ and BP exercises successively in an alternating manner (Figure 1). The same inter-set recovery was established between sets of the same exercise (3 min). For the purpose of the experimental study, both groups completed four testing sessions (72–96 h apart), increasing the relative load by 5% throughout each session, and using the same (1) relative loads (55–70% 1RM) for each exercise, (2) magnitude of velocity loss (%VL) in each training set (15% and 20% for SQ and BP, respectively), (3) exercise order (SQ followed by BP), (4) number of sets per exercise (three); and (5) inter-set recovery (3 min).

Two weeks prior to the first testing session, all subjects underwent 3 familiarization sessions (≥48 h apart) to be instructed in the execution technique of each exercise (e.g., initial and final position, and lifting the load at maximal intended velocity). During this period, the subjects also performed a progressive loading test in the SQ and BP exercises for the estimation of the 1RM. Subjects were required to refrain from any other type of strenuous physical activity, exercise training, or sports competition for the duration of the investigation. Every testing session was conducted in a research laboratory under the direct supervision of two experienced investigators, at the same time of the day (±1.5 h) for each participant and under similar environmental conditions (21–24 °C and 55–62% humidity).

### 2.2. Subjects

A total of 19 physically active young men (age: 24.0 ± 5.0 years, body mass: 73.1 ± 9.5 kg, height: 1.73 ± 0.08 m) volunteered to participate in this study. Subjects were moderately strength-trained (SQ 1RM: 93.6 ± 19.1 kg, BP 1RM: 72.4 ± 12.4 kg), ranging from 6 months to 3 years (1–3 sessions per week) of RT experience, and had been injury-free for at least 6 months prior to the study. All subjects declared not to have any type of drug, dietary supplement, or medication that may alter physical performance. After an initial evaluation, subjects were matched according to their relative strength ratio (1RM/body mass) in the SQ and BP exercises and then randomly assigned into two groups depending on how the scheduled sets were performed between exercises: traditional-set (TS, *n* = 9) or alternating-set (AS, *n* = 10) groups. The study was conducted according to the Declaration of Helsinki and was approved by the local Research Ethics Committee. Prior to participation, all subjects signed a written consent form after being informed of the risks, purpose, and experimental procedures of the study.

### 2.3. Testing Procedures

Descriptive characteristics of the loads used for each testing session are reported in Table 1 (SQ exercise) and Table 2 (BP exercise). All sessions were performed on a Smith machine (Multipower, Technogym), and repetitions were measured and recorded using a linear velocity transducer (T-Force Dynamic Measurement System; Ergotech Consulting Ltd., Murcia, Spain). A complete analysis of this device’s reliability is reported elsewhere [19,20]. The velocity measures reported in this study corresponded to the mean velocity of the propulsive phase (i.e., MPV), defined as the portion of the concentric action during which the measured acceleration is greater than acceleration due to gravity (−9.81 m·s^−1^) [20].

All subjects completed four testing sessions (72–96 h apart) with increased loading throughout each session (55%, 60%, 65%, and 70% 1RM), performing the SQ before the BP exercise. Training variables, including relative loads (55–70% 1RM), number of sets (three), magnitude of %VL within the set (15% and 20% for the SQ and BP exercises, respectively), and inter-set recovery (3 min), were identical for the two experimental conditions during each testing session. The only difference between both groups was the set configuration performed between exercises: traditional or alternating manner. This range of moderate relative intensities (i.e., 55–70% 1RM) was used because they are considered as adequate training loads for improving performance in high-speed actions (i.e., jumping and sprinting) [21,22]. During each testing session, subjects received immediate velocity feedback while being encouraged to perform each repetition during the concentric action at maximal intended velocity (i.e., as fast as possible). Preceding each session, subjects of both groups conducted a general standardized warm-up, consisting of 5 min of jogging at a self-selected easy pace, joint mobilization exercises and dynamic stretching, and a specific warm-up protocol of (i) one set of 8 repetitions at moderate velocity (against 25 kg for SQ and BP), (ii) one set of 5 repetitions at high velocity (against 25 kg for SQ and BP), and (iii) one set of 2–3 repetitions at maximal intended velocity (against the proposed absolute load that best matched the scheduled target MPV for SQ and BP), with a 2 min inter-set rest.

Since this study was conducted on a VBRT approach, individualization of the relative load (%1RM) for each testing session was determined from the general load–velocity relationship for SQ [23] and BP [24]. Thereby, the target velocity to be attained in the first (usually the fastest) repetition of the first set of each session was used as an indicator of the relative load for all subjects. Consequently, before starting the first set in each testing session, adjustments in the proposed load (kg) were individually made to match the scheduled target MPV (±0.03 m·s^−1^) associated with the %1RM that was set for the specific session. A range of 0.03 m·s^−1^ was used since it has recently been shown that the smallest detectable change in MPV when using the T-Force System is 0.03 m·s^−1^ [19]. Once the load (kg) was adjusted, it was maintained for the three sets. The volume (number of repetitions) to be performed in each exercise set was objectively determined by means of the percentage of VL attained in the set [25] so that each set was finished as soon as the prescribed %VL limit was achieved regardless of the number of repetitions completed by each participant [18,26,27]. Fixed magnitudes of intra-set VL were established for all sessions to provide a homogeneous level of effort or fatigue across the subjects at the end of each set [17]. Moderate %VL (i.e., 15% vs. 20% for SQ and BP, respectively) was used because it represents a moderate degree of fatigue incurred in the set (i.e., less than half of the maximum number of repetitions that can be completed in a set to failure) [17]. Different magnitudes of %VL for each exercise were used to match the same percentage of repetitions per set completed with respect to the maximum possible repetitions against each relative load [25].

### 2.4. Execution Technique of the SQ and BP Exercises

A detailed description of the progressive loading test protocol for both exercises has been provided elsewhere [23,28], and the execution technique was exactly reproduced throughout the study. For the full SQ exercise, subjects started from the upright position with the knees and hips fully extended, stance approximately shoulder-width apart, and the barbell resting across the back at the level of the acromion. Subjects descended in a continuous motion until the posterior thighs and calves contacted each other, then immediately reversed the motion and ascended back to the upright position. Subjects were required to execute each repetition without any pause between the eccentric and concentric action, whereas the eccentric phase was performed at controlled velocity (range: 0.50–0.65 m·s^−1^). Subjects were also required to keep their feet in contact with the ground (that is, without jumping), though their heels could lift slightly.

For the BP exercise, subjects were also required to execute the eccentric phase of each repetition at controlled velocity, and a momentary pause (~1 s) of the barbell on the lateral supports of the Smith machine (1–2 cm above the chest) was imposed between the eccentric and concentric phase to minimize the contribution of the stretch–shortening cycle and allow more reliable and consistent measures of movement velocity [28]. Following the momentary pause of the barbell, subjects were instructed to push it at maximal intended velocity. In this exercise, the feet were placed on the bench to avoid lumbar arching, and hands gripped the barbell slightly wider (5–7 cm) than shoulder width. The position of the bench was adjusted so that the vertical projection of the barbell coincided with the intermammary line of each participant.

### 2.5. Statistical Analysis

Values are reported as means and standard deviations (SD). The normality of distribution of the variables was examined with the Shapiro–Wilk test, and the homogeneity of variance across groups (TS vs. AS) was verified using Levene’s test. Differences in testing variables (MPV of the first repetition, percentage of intra-set VL, and number of repetitions performed at different velocity ranges) between groups for each testing session were examined using a Student’s t-test for independent variables. Similarly, between-set inter-group differences in MPV and number of repetitions were assessed using a 3 (sets: 1° vs. 2° vs. 3°) × 2 (group: TS vs. AS) repeated measures factorial ANOVA with Bonferroni’s adjustment for every session (i.e., relative load, %1RM). Statistical significance was accepted at *p* < 0.05. Null hypothesis tests were performed using SPSS software version 25.0 (SPSS, Chicago, IL, USA).

## 3. Results

Differences between TS and AS in the selected neuromuscular performance variables for SQ and BP exercises are displayed in Table 1, Table 2, Table 3, Table 4, Table 5 and Table 6.

### 3.1. Velocity-Based Testing Variables

Both the fastest MPV (i.e., the relative load, %1RM) and the %VL actually performed over the three sets closely matched those scheduled for each testing session and exercise in both groups (Table 1 and Table 2). Consequently, no significant differences were observed in the average %VL attained over the three sets between groups during the entire testing sessions in the SQ (16.6 ± 1.4% and 16.8 ± 1.5% for TS and AS groups, respectively) and BP exercise (21.6 ± 1.7% and 21.4 ± 2.0% for TS and AS groups, respectively).

### 3.2. Execution Velocity (MPV_FIRST_)

No significant differences between groups were observed for the MPV of the first repetition (average of the three sets) in any testing session and exercise (Table 1 and Table 2). Therefore, subjects of the TS and AS groups executed the repetitions at the same average MPV of all testing sessions (0.94 ± 0.03 m·s^−1^ vs. 0.93 ± 0.03 m·s^−1^ in SQ and 0.73 ± 0.02 m·s^−1^ vs. 0.72 ± 0.02 m·s^−1^ in BP, respectively).

### 3.3. Volume (Number of Repetitions)

No significant differences in the number of repetitions per set (average of the three sets) were observed for any relative load and exercise (Table 1 and Table 2). Comparisons of the total number of average repetitions per set in the SQ exercise revealed that the TS group performed a not statistically significant higher number of repetitions than the AS group (30.3 ± 2.8 vs. 24.4 ± 1.7, respectively). However, both experimental groups completed a very similar total number of average repetitions per set in the BP exercise (26.4 ± 2.3 vs. 26.0 ± 1.9 for TS and AS groups, respectively).

With respect to the number of repetitions performed at different velocity ranges in the SQ exercise (Table 3), the TS group completed a greater number of repetitions at faster velocities (MPV > 0.90–1.10 m·s^−1^) and a higher number of total repetitions than the AS group (90.8 ± 27.0 vs. 73.1 ± 18.1, *p* < 0.05, respectively). For the BP exercise, there were no significant differences between groups in the number of repetitions completed, except for the velocity range of 0.80–0.90 m·s^−1^ (Table 4). Both groups performed a similar number of total repetitions (75.1 ± 23.9 vs. 73.5 ± 12.9 for TS and AS groups, respectively) in the whole spectrum of velocity ranges.

### 3.4. Between-Set Execution Velocity and Number of Repetitions

Comparisons of the fastest MPV (i.e., %1RM) between sets showed no significant differences between groups (TS vs. AS) for any relative load and exercise (Table 5 and Table 6). Between-group comparisons of the number of repetitions performed between sets revealed significant differences (*p* < 0.05) in the BP exercise with the loads corresponding to 55% (first set), 60% (first set), and 65% (third set) 1RM.

### 3.5. Total Training Time per Session

Between-group differences in total training time completed per session, including the standardized warm-up, revealed significantly shorter (*p* < 0.001) total workout duration for AS group (23.3 ± 2.2 min) than for TS group (42.2 ± 3.1 min).

## 4. Discussion

The purpose of the present study was to compare the acute effect on bar execution velocity and volume of two different set configurations (TS vs. AS) in the SQ and BP exercises. To the best of our knowledge, this is the first study comparing traditional- vs. alternating-set configurations: (1) using paired exercises involving the upper- and lower-limbs muscle groups (SQ and BP exercises); (2) applying the VBRT methodology to set the relative load (%1RM) and volume (%VL in the set); (3) inducing a moderate-to-low degree of fatigue in each set (i.e., % intra-set VL), that is, ending each set well ahead of reaching muscle failure; and (4) measuring the neuromuscular performance based on execution velocity (i.e., fastest MPV) and volume (i.e., number of repetitions per set). The main finding of the current investigation was that the AS group performed a similar execution velocity and number of repetitions per set to the TS group over the different testing sessions, confirming our hypothesis. Therefore, considering that the total workout duration was significantly shorter (approximately half) for the AS group, our results suggest this RT configuration could constitute a more time-efficient training method than the TS configuration since it allows optimizing training time without compromising acute neuromuscular performance over the training session. This occurs, at least, when both combined exercises (SQ and BP) are conducted with moderate relative loads (55–70% 1RM) and a moderate degree of fatigue (15%VL and 20%VL for the SQ and BP exercises, respectively).

The AS configuration applied in this study resulted in a similar average MPV of the three sets and MPV between sets in the SQ and BP exercises than the traditional condition (Table 1, Table 2, Table 5 and Table 6). This was observed with every relative load used in the present study (i.e., 55, 60, 65, and 70% 1RM). These results appear to indicate that, compared to the TS, the AS configuration did not negatively affect neuromuscular performance at the beginning of each set when a moderate degree of fatigue (i.e., 15–20% intra-set VL) was induced in the previous exercise. On the other hand, both groups (TS and AS) completed a similar number of repetitions per set in the BP exercise for every relative load (Table 2) and in most of the different velocity ranges (Table 4), indicating that training volume was not significantly impaired in the AS condition despite being alternated with the SQ exercise in the same set. However, comparisons between groups in the SQ exercise revealed that the TS group performed a higher number of total repetitions during all sessions than those completed by the AS group (Table 1 and Table 3), especially at faster velocities. This fact could be explained because of the lower degree of fatigue experienced in the TS group by not interspersing the BP exercise during the inter-set recovery intervals of the SQ exercise. However, it is unknown if this acute impairment with respect to the number of repetitions in the AS condition would affect strength development following a training intervention.

The effects of consecutive completion of two different exercises targeting the same or antagonist muscle groups followed by a recovery period (also called super-set and paired-set training) have been previously addressed [5,6,7]. These techniques represent training strategies observed in the real-world setting that have been recognized for saving training time with respect to a traditional approach by reducing the time spent during passive rest [8,9]. However, the paucity of cross-over studies examining upper- and lower-limb exercise pairings (e.g., SQ and BP) on variables related to mechanical and neuromuscular performance (i.e., velocity, power, force) makes a direct comparison with our results difficult. Some evidence has suggested that it is unlikely that upper-body exercises performed in an AS condition directly affect the central drive to the lower-body muscles involved in the SQ exercise [2]. In this study, the traditional protocol only consisted of SQ sets (4 × 80% 1RM, 3 min inter-set recovery time), whereas the AS condition performed BP and bench pull exercises between SQ sets with between-exercise rest of 50 s, resulting in approximately 3 min inter-set recovery time between SQ sets. For all exercises, four repetitions were completed for Sets 1 to 3, whereas the fourth set was performed to concentric failure. The main finding of this study was that performing upper-body multi-joint exercises during SQ rest intervals (i.e., AS condition) impaired the number of SQ repetitions to failure and volume-equated average power in the fourth set compared to the TS condition. The results of our study are consistent with those of this previous study since, in the first three sets, not performed to muscle failure, an SQ acute performance impairment in the AS condition was not observed. However, this study was not based on a VBRT approach, did not report the effect on BP and bench pull performance, and only examined the effect using 80% 1RM loads. Similarly, a previous study showed that performing AS between SQ and BP (3 × 10 repetitions performed at maximal velocity, 2 min inter-set recovery, and 65% of 3RM load) had significant and greater reductions in mean velocity and power between sets in the BP exercise compared to bent-over row and BP paired sets and the traditional condition (where the BP was performed alone) [13]. However, the discrepancies with our results could be explained by differences in the configuration of the RT sessions analyzed in both studies (e.g., inter-set recovery time, matching of the relative load and number of repetitions per set, time elapsed between exercises). In line with our results, research has also found that the inclusion of lower-body single-joint exercises in a circuit training fashion (bench press + leg extensions + ankle extensions) did not significantly affect BP bar velocity, power, and number of repetitions performed to volitional fatigue (5 × 6RM, 3 min active rest between sets) compared to a TS condition [29]. However, this study did not report differences between training conditions on the lower-body exercise performance, and, additionally, sets were performed to muscle failure. Finally, Robbins, Young, Behm, and Payne (2010) [6] compared the acute effects of performing TS versus AS between BP throw and bench pull on peak velocity, peak power, bench press thrown height, volume load per set, and session and electromyographic activity (3 sets; 4 min inter-set recovery time). No differences were found in any of the variables studied between both training conditions, so the main conclusion of this study was that performing agonist–antagonist paired sets would appear to be an effective training method with respect to efficiency and maintenance of neuromuscular performance. Although these findings are similar to the present study, it should be considered that the exercises used did not involve opposite limbs (i.e., upper- and lower-body muscle groups), and the bench pull was performed to muscle failure.

The findings of this investigation may serve as a practical guideline to effectively implement AS into RT programs aimed at maintaining neuromuscular performance (i.e., execution velocity and repetitions per set) over sets, especially for those individuals with time constraints to perform sessions combining upper- and lower-body exercises. Based on our results, performing AS of two resistance exercises should not have a substantially detrimental effect on neuromuscular performance during consecutive sets, in comparison with a traditional approach, provided that certain rules are respected: (i) paired/coupled exercises must involve different body segments or agonist muscles (i.e., upper- and lower-body muscle groups); (ii) a moderate-to-low degree of fatigue in each training set should be achieved in both paired exercises (i.e., ≤15–20% intra-set VL), that is, performing sets ending well ahead of reaching muscle failure; and (iii) the inter-set recovery time for each exercise has to be long enough to allow a complete or almost complete neuromuscular recovery (~3 to 5 min).

The relatively small sample size and the heterogeneity in the strength levels among participants must be considered as the major limitation of the present study. In addition, it would be necessary to analyze the chronic effects that a training program with these characteristics (i.e., relative loads and level of fatigue (%VL) incurred in the set) could have on neuromuscular performance and strength gains.

## 5. Conclusions

Training sessions performing AS between SQ and BP exercises with moderate loads and achieving a moderate degree of fatigue in both exercises resulted in similar neuromuscular performance (i.e., execution velocity and number of repetitions per set), but in a more time-efficient manner, than the traditional approach.

## Figures and Tables

**Figure 1 sports-10-00110-f001:**
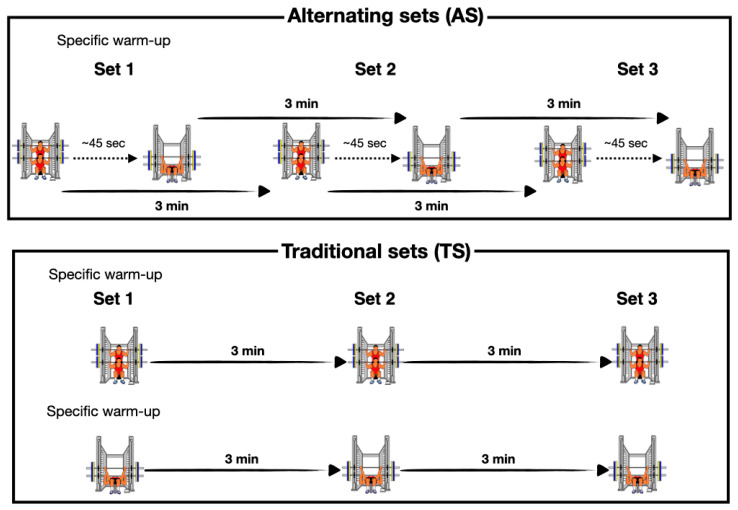
Schematic representation of the training protocol for alternating-set (**top panel**) and traditional-set configurations (**bottom panel**). Note: A ~45 s rest interval time between the completion of every set of SQ and the subsequent set of BP was implemented for the alternating-set group. This interval time was used by the assistants to adjust individually the appropriate BP absolute load and place the bench in the Smith machine.

**Table 1 sports-10-00110-t001:** Descriptive characteristics of the velocity-based squat testing protocol performed by both experimental groups.

Scheduled	Session 1	Session 2	Session 3	Session 4	Average
Target MPV (m·s^−1^)	1.07	1.00	0.92	0.84	0.96
	(~55% 1RM)	(~60% 1RM)	(~65% 1RM)	(~70% 1RM)	(62.5%)
Sets x VL (%)	3 × 15%	3 × 15%	3 × 15%	3 × 15%	3 × 15%
Actually performed					Average
MPV_FIRST_ (m·s^−1^)					
TS	1.06 ± 0.03	0.98 ± 0.03	0.89 ± 0.03	0.82 ± 0.02	0.94 ± 0.03
	(~56.2% 1RM)	(~61.5% 1RM)	(~67.3% 1RM)	(~71.3% 1RM)	(64.0% 1RM)
AS	1.04 ± 0.03	1.00 ± 0.04	0.89 ± 0.01	0.81 ± 0.03	0.93 ± 0.03
	(~57.6% 1RM)	(~60.6% 1RM)	(~67.2% 1RM)	(~72.0% 1RM)	(64.3% 1RM)
Intra-set VL (%)					Average
TS	16.8 ± 1.3	17.3 ± 2.1	16.0 ± 0.9	16.2 ± 1.2	16.6 ± 1.4
AS	17.8 ± 1.2	17.6 ± 2.4	15.5 ± 1.1	16.3 ± 1.2	16.8 ± 1.5
Reps per set (#)					Total
TS	11.1 ± 2.5	8.4 ± 3.7	6.1 ± 2.9	4.7 ± 1.1	30.3 ± 2.8
AS	8.0 ± 1.8	6.6 ± 2.8	5.4 ± 1.7	4.2 ± 1.1	24.4 ± 1.7

*Note:* Data are mean ± SD. MPV: mean propulsive velocity attained against the intended load (%1RM); VL: Velocity loss; Reps per set: number of repetitions performed; MPV_FIRST_: mean propulsive velocity of the fastest (usually first) repetition in the set. The actual MPV, velocity losses and repetitions per set reported are the mean of the three sets. TS: Traditional-set group. AS: Alternating-set group.

**Table 2 sports-10-00110-t002:** Descriptive characteristics of the velocity-based bench-press testing protocol performed by both experimental groups.

Scheduled	Session 1	Session 2	Session 3	Session 4	Average
Target MPV (m·s^−1^)	0.87	0.78	0.70	0.62	0.74
	(~55% 1RM)	(~60% 1RM)	(~65% 1RM)	(~70% 1RM)	(62.5%)
Sets × VL (%)	3 × 20%	3 × 20%	3 × 20%	3 × 20%	3 × 20%
Actually performed					Average
MPV_FIRST_ (m·s^−1^)					
TS	0.85 ± 0.02	0.77 ± 0.03	0.69 ± 0.02	0.61 ± 0.01	0.73 ± 0.02
	(~56.0% 1RM)	(~60.4% 1RM)	(~65.6% 1RM)	(~70.7% 1RM)	(63.2%)
AS	0.84 ± 0.03	0.76 ± 0.02	0.68 ± 0.02	0.60 ± 0.02	0.72 ± 0.02
	(~56.3% 1RM)	(~61.2% 1RM)	(~66.2% 1RM)	(~71.0% 1RM)	(63.7%)
Intra-set VL (%)					Average
TS	23.2 ± 1.6	21.8 ± 1.8	20.4 ± 1.2	21.1 ± 2.3	21.6 ± 1.7
AS	22.2 ± 2.5	22.7 ± 1.9	20.5 ± 1.9	20.5 ± 1.8	21.4 ± 2.0
Reps per set (#)					Total
TS	9.7 ± 3.5	6.9 ± 2.1	5.6 ± 1.3	4.3 ± 0.7	26.4 ± 2.3
AS	8.4 ± 1.5	7.8 ± 1.7	5.6 ± 1.2	4.3 ± 0.8	26.0 ± 1.9

*Note:* Data are mean ± SD. MPV: mean propulsive velocity attained against the intended load (%1RM); VL: Velocity loss; Reps per set: number of repetitions performed; MPV_FIRST_: mean propulsive velocity of the fastest (usually first) repetition in the set. The actual MPV, velocity losses, and repetitions per set reported are the mean of the three sets. TS: Traditional-set group. AS: Alternating-set group.

**Table 3 sports-10-00110-t003:** Number of repetitions performed in each velocity range and total number of repetitions completed by both training groups in squat.

MPV (m·s^−1^)	TS	AS
<0.3	0.0 ± 0.0	0.0 ± 0.0
>0.3–0.4	0.0 ± 0.0	0.0 ± 0.0
>0.4–0.5	0.0 ± 0.0	0.0 ± 0.0
>0.5–0.6	0.0 ± 0.0	0.0 ± 0.0
>0.6–0.7	3.2 ± 2.0	3.1 ± 1.7
>0.7–0.8	14.3 ± 3.5	15.0 ± 5.1
>0.8–0.9	27.1 ± 9.3	24.5 ± 8.3
>0.9–1.0	31.6 ± 15.5	22.0 ± 7.1 *
>1.0–1.1	14.1 ± 9.2	8.5 ± 3.4 *
>1.1	0.6 ± 1.3	0.1 ± 0.3
Total reps	90.8 ± 27.0	73.1 ± 18.1 *

*Note:* Data are mean ± SD. The experimental groups performed different set configurations: TS (*n* = 9), AS (*n* = 10). Statistically significant differences with respect to TS group (* *p* < 0.05). Abbreviations: TS: Traditional-set experimental group; AS: Alternating-set experimental group; MPV: mean propulsive velocity; Reps: number of repetitions performed.

**Table 4 sports-10-00110-t004:** Number of repetitions performed in each velocity range and total number of repetitions completed by both training groups in bench press.

MPV (m·s^−1^)	TS	AS
<0.3	0.0 ± 0.0	0.0 ± 0.0
>0.3–0.4	0.0 ± 0.0	0.2 ± 0.6
>0.4–0.5	2.6 ± 0.7	2.7 ± 1.5
>0.5–0.6	15.7 ± 3.9	17.6 ± 4.5
>0.6–0.7	26.7 ± 6.9	28.3 ± 5.8
>0.7–0.8	24.6 ± 12.2	22.2 ± 5.5
>0.8–0.9	9.9 ± 3.6	6.8 ± 2.6 *
>0.9–1.0	0.0 ± 0.0	0.3 ± 0.9
>1.0–1.1	0.0 ± 0.0	0.0 ± 0.0
>1.1	0.0 ± 0.0	0.0 ± 0.0
Total reps	75.1 ± 23.9	73.5 ± 12.9

*Note:* Data are mean ± SD. The experimental groups performed different set configurations: TS (*n* = 9), AS (*n* = 10). Statistically significant differences with respect to TS group (* *p* < 0.05). Abbreviations: TS: Traditional-set experimental group; AS: Alternating-set experimental group; MPV: mean propulsive velocity; Reps: number of repetitions performed.

**Table 5 sports-10-00110-t005:** Execution velocity and number of repetitions per set performed by both experimental groups in the full-squat exercise.

Session (#)	1			2			3			4		
Target MPV (m·s^−1^)	1.07 (~55% 1RM)			1.00 (~60% 1RM)			0.92 (~65% 1RM)			0.84 (~70% 1RM)		
Set (#)	1	2	3	1	2	3	1	2	3	1	2	3
MPV_FIRST_ (m·s^−1^)												
TS group	1.09 ± 0.03	1.05 ± 0.03	1.06 ± 0.04	0.99 ± 0.03	0.98 ± 0.05	0.97 ± 0.03	0.91 ± 0.02	0.88 ± 0.04	0.87 ± 0.04	0.84 ± 0.02	0.81 ± 0.03	0.81 ± 0.03
AS group	1.09 ± 0.03	1.04 ± 0.04	1.00 ± 0.06	1.01 ± 0.02	0.99 ± 0.06	0.98 ± 0.07	0.91 ± 0.02	0.88 ± 0.02	0.88 ± 0.02	0.84 ± 0.01	0.80 ± 0.05	0.79 ± 0.03
Reps per set (#)												
TS group	10.4 ± 3.8	12.1 ± 2.8	10.8 ± 3.9	7.9 ± 3.9	8.6 ± 3.9	8.7 ± 5.1	6.8 ± 2.4	5.2 ± 2.0	6.3 ± 4.6	5.0 ± 1.7	4.6 ± 1.2	4.6 ± 1.0
AS group	7.9 ± 2.4	8.1 ± 2.8	8.2 ± 2.7	7.0 ± 2.5	6.0 ± 0.9	7.2 ± 5.5	6.0 ± 2.2	5.6 ± 1.9	4.7 ± 1.6	4.3 ± 1.2	4.3 ± 1.3	3.9 ± 1.3

*Note:* Data are mean ± SD. MPV: mean propulsive velocity attained against the intended load (%1RM); Reps per set: number of repetitions performed; MPVFIRST: mean propulsive velocity of the fastest (usually first) repetition in each set. TS: Traditional-set group. AS: Alternating-set group.

**Table 6 sports-10-00110-t006:** Execution velocity and number of repetitions per set performed by both experimental groups in the bench-press exercise.

Session (#)	1			2			3			4		
Target MPV (m·s^−1^)	0.87 (~55% 1RM)			0.78 (~60% 1RM)			0.70 (~65% 1RM)			0.62 (~70% 1RM)		
Set (#)	1	2	3	1	2	3	1	2	3	1	2	3
MPV_FIRST_ (m·s^−1^)												
TS group	0.87 ± 0.02	0.85 ± 0.03	0.83 ± 0.02	0.78 ± 0.03	0.77 ± 0.03	0.78 ± 0.04	0.69 ± 0.02	0.69 ± 0.02	0.69 ± 0.03	0.62 ± 0.02	0.60 ± 0.02	0.61 ± 0.02
AS group	0.87 ± 0.04	0.84 ± 0.03	0.82 ± 0.04	0.78 ± 0.02	0.75 ± 0.02	0.75 ± 0.04	0.70 ± 0.02	0.67 ± 0.03	0.66 ± 0.03	0.62 ± 0.02	0.60 ± 0.02	0.59 ± 0.03
Reps per set (#)												
TS group	10.2 ± 4.2	9.8 ± 3.2	9.1 ± 3.1	7.6 ± 3.1	6.7 ± 1.9	6.3 ± 1.7	5.3 ± 1.1	5.6 ± 1.7	5.8 ± 1.4	4.4 ± 0.9	4.4 ± 0.9	4.1 ± 0.9
AS group	8.6 ± 2.0 *	8.6 ± 1.9	7.9 ± 1.7	8.3 ± 2.3 *	8.3 ± 1.4	6.8 ± 1.9	6.2 ± 1.1	5.3 ± 1.5	5.3 ± 1.6 *	4.6 ± 1.2	4.3 ± 0.9	3.9 ± 0.9

*Note:* Data are mean ± SD. MPV: mean propulsive velocity attained against the intended load (%1RM); Reps per set: number of repetitions performed; MPVFIRST: mean propulsive velocity of the fastest (usually first) repetition in each set. TS: Traditional-set group. AS: Alternating-set group. Statistically significant differences with respect to: * TS group (*p* < 0.05).

## Data Availability

The study did not report any data.

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
