# Peer review of "Acute Effect of Upper-Lower Body Super-Set vs. Traditional-Set Configurations on Bar Execution Velocity and Volume"

_sports, 2022, doi:10.3390/sports10070110_

Round 1

Reviewer 1 Report

The manuscript is well written with detailed description of the methods and proper interpretation of the results. The organization and subsections are also appropriate. The manuscript is structured and presented in a reader-friendly manner.

The manuscript should contain at least 1 figure and 6 tables. However, I can’t find them in it. Without them, I may not be able to accurately understand the experimental process and analyze the validity of these data.

Author Response

The figure and tables are inserted in the new manuscript version. Spell checking has been made.

Reviewer 2 Report

General comments

The authors have clearly stated that the purpose of the study was to compare the effect on bar execution velocity and volume (i.e., repetitions per set) of traditional vs. alternating set configuration in the full-squat and bench-press exercises using a velocity-based resistance training approach. The paper is well-written and easy to follow. In my opinion, it adds considerable value to the current literature, since resistance training is a very hot topic in exercise science affecting numerous health and performance indicators among athletic, general, and special populations. This study can enhance future attempts in similar research area in order to investigate more specific pathways between velocity-based resistance training configuration and acute as well as chronic effects in various populations. However, I have highlighted a few suggestions and concerns in my specific comments section (below) that need to be addressed before considering whether this work should be published or not.

Specific comments

ABSTRACT

  • There is no abstract in the submitted manuscript. Please address this issue accordingly.

  1. INTRODUCTION

- Nice work from the authors. I suggest adding a few statements to connect the topic with the current state of the health and fitness industry, and more specifically with the top relevant trends in this sector, since resistance training with free-weights is one of the most attractive trends in Europe and worldwide among exercise professionals. Thus, consider citing the following studies.

References:

  1. Kercher et al. 2022 Fitness Trends from Around the Globe. ACSMs Health Fit J 2022; 26(1): 21–37.
  2. Batrakoulis A. European Fitness Trends for 2020. ACSMs Health Fit J 2019; 23(6): 28–35.

  1. DISCUSSION

  • Please provide a short paragraph, presenting the major limitations of the present study at the end of the discussion section.

  1. CONCLUSIONS

  • Please add more practical implications in this section.

Author Response

Abstract has been added to the manuscript.

Introduction

The authors do not find necessary to cite and add the studies suggested by the reviewer.

Discussion

A short paragraph, presenting the major limitations of the present study at the end of the Discussion section is provided.

Conclusions

Practical implications were already listed at the end of the Discussion section: ...performing AS of two resistance exercises should not have substantial detrimental effect on neuromuscular performance during consecutive sets, in comparison with a traditional approach, provided that certain rules are respected: i) paired/coupled exercises must involve different body segments or agonist muscles (i.e., upper- and lower-body muscle groups), ii) a moderate-to-low degree of fatigue in each training set should be achieved in both paired exercises (i.e., ≤15-20% intra-set VL), that is, performing sets ending well ahead of reaching muscle failure, and iii) the inter-set recovery time for each exercise has to be long enough to allow a complete or almost complete neuromuscular recovery (~3 to 5 min).

This practical implications paragraph could be moved to the Conclusion section if needed.
